# Ursolic Acid Inhibits Collective Cell Migration and Promotes JNK-Dependent Lysosomal Associated Cell Death in Glioblastoma Multiforme Cells

**DOI:** 10.3390/ph14020091

**Published:** 2021-01-26

**Authors:** Gillian E. Conway, Deimante Zizyte, Julie Rose Mae Mondala, Zhonglei He, Lorna Lynam, Mathilde Lecourt, Carlos Barcia, Orla Howe, James F. Curtin

**Affiliations:** 1School of Food Science and Environmental Health, Technological University Dublin, D01 HV58 Dublin, Ireland; deimante.zizyte@tudublin.ie (D.Z.); julie.mondala@tudublin.ie (J.R.M.M.); zhonglei.he@tudublin.ie (Z.H.); lornal95@live.ie (L.L.); lecourt.mathilde31@gmail.com (M.L.); 2Environmental Sustainability and Health Institute (ESHI) and FOCAS Research Institute, Technological University Dublin, D08 CKP1 Dublin, Ireland; orla.howe@tudublin.ie; 3In-Vitro Toxicology Group, Institute of Life Science, Swansea University Medical School, Swansea University, Singleton Park, Swansea SA2 8PP, UK; 4Institut de Neurociències, Department of Biochemistry and Molecular Biology, School of Medicine, Universitat Autònoma de Barcelona, 08193 Bellaterra, Spain; carlos.barcia@uab.es; 5School of Biological Sciences and Health Sciences, Technological University Dublin, D08 NF82 Dublin, Ireland

**Keywords:** ursolic acid, cell death, migration, lysosomes, nutraceuticals

## Abstract

Ursolic acid (UA) is a bioactive compound which has demonstrated therapeutic efficacy in a variety of cancer cell lines. UA activates various signalling pathways in Glioblastoma multiforme (GBM) and offers a promising starting point in drug discovery; however, understanding the relationship between cell death and migration has yet to be elucidated. UA induces a dose dependent cytotoxic response demonstrated by flow cytometry and biochemical cytotoxicity assays. Inhibitor and fluorescent probe studies demonstrate that UA induces a caspase independent, JNK dependent, mechanism of cell death. Migration studies established that UA inhibits GBM collective cell migration in a time dependent manner that is independent of the JNK signalling pathway. Cytotoxicity induced by UA results in the formation of acidic vesicle organelles (AVOs), speculating the activation of autophagy. However, inhibitor and spectrophotometric analysis demonstrated that autophagy was not responsible for the formation of the AVOs. Confocal microscopy and isosurface visualisation determined co-localisation of lysosomes with the previously identified AVOs, thus providing evidence that lysosomes are likely to be playing a role in UA induced cell death. Collectively, our data identify that UA rapidly induces a lysosomal associated mechanism of cell death in addition to UA acting as an inhibitor of GBM collective cell migration.

## 1. Introduction

Glioblastoma multiforme (GBM) is a highly malignant grade IV astrocytoma. It is considered to be the most biologically aggressive brain tumour and is associated with a poor post diagnosis survival rate of about 1 year [1]. GBMs are a heterogeneous mixture of cells that display varying degrees of cellular and nuclear polymorphism, making it difficult to manage clinically. The current standard of care for GBM is maximal surgical resection followed by radiotherapy and concomitant and adjuvant chemotherapy using Temozolomide (TMZ) [2]. The efficacy of Temozolomide and radiotherapy has been limited due to treatment toxicity, therapeutic resistance and failure to successfully remove all surrounding tumour cells during surgical resection [3], therefore resulting in a high chance of tumour recurrence. Consequently, survival rates have remained relatively stagnant over the past 30 years [4]. In order to improve clinical outcomes for patients undergoing therapy, there is a need for the development of new therapeutic strategies that investigate alternative compounds that possess cytotoxic activity with the ability to inhibit tissue invasion.

In an effort to combat these issues, research groups have identified numerous naturally derived bioactive compounds (NDBC), known as phytochemicals, that demonstrate anticancer activity when used alone or in combination for the treatment of cancer [5]. Phytochemicals with anti-cancer properties are being assessed as precursors for new chemotherapeutic agents [6,7,8,9,10]. Ursolic acid (UA), a ursane-type pentacyclic triterpenoid acid, is a primary component of the thin waxy coating in popular western foods such as cranberries, apples and olives [11,12,13,14]. UA, along with other triterpenoids, has been used in Chinese Traditional Medicine for centuries to treat cancer and inflammatory disorders [15,16]. In recent years, UA has been shown both in vitro and in vivo to demonstrate anti-proliferative, anti-migratory, and anti-inflammatory properties and enhance the efficacy and sensitivity of chemotherapeutics in a variety of cancer types, including GBM [17,18]. An advantage of many naturally derived compounds such as UA is that they have the capacity to elicit a biological response while being nontoxic at low doses compared to commonly used chemotherapeutics [19], making them an ideal candidate for combinational therapy to act as a chemosensitizer. In a recent study, UA was shown to increase sensitivity to TMZ in TMZ-resistant GBM cells (T98G and LN18) in vitro [20]. 

UA offers a promising strategy for the treatment of GBM; however, the mechanisms of action have not been fully elucidated in the current literature. In GBM models, low concentrations of UA and natural esters of UA have been shown to inhibit the proliferation of U-87 and SF-295 cells [21,22] and prevent invasion induced by IL1β and TNFα in C6 rat glioma cells [23]. Additionally in other GBM models, UA has been shown to induce necrosis in TMZ resistant DBTRG-05MG cells [12], cell cycle arrest and autophagy in U-87 cells [24], apoptosis in U251 cells [13], and both anti-proliferative and apoptotic effects in C6 rat glioma cells in vitro [14]. It is postulated that the mechanism of cell death is cell line dependent and is likely to alter depending on the genetic mutations of the cell line. In recent years, the Nomenclature Committee on Cell Death (NCCD) has established guidelines for the interpretation cell death data [25]. In 2018, they updated the guidelines to include lysosome-dependant cell death (LDCD) and autophagy dependent cell death (ADCD). LDCD is defined as a form of regulated cell death identified lysosomal membrane permeabilization (LMP), resulting from the release of Cathepsins with or without mitochondrial outer membrane permeabilization. Autophagy, a traditionally cytoprotective process, is also recognised as a form of regulated cell death by blocking the cells cytoprotective mechanisms, therefore resulting in cell death. However, due to the involvement of lysosomes in the autophagic pathway (i.e., fusion of the lysosome with the autophagosome), LDCD can often be overlooked. 

Several key questions have surfaced following observations that UA activates a variety of signalling pathways in GBM. This study carries out a direct comparison of UA with existing GBM therapies to investigate the role of UA in cell death and GBM migration with the aim of elucidating the molecular mechanisms involved in UA associated cytotoxicity.

## 2. Results

### 2.1. UA Induced Cytotoxicity Results in Rapid Mitochondrial Membrane Depolarisation

UA was found to induce rapid cytotoxicity in U-251 MG cells within 24 h of treatment. The rapid onset of cell death was accompanied by a characteristic, steep Hill slope, with no significant difference in IC_50_ values determined when cells were treated with UA for 24 h, 48 h or 6 days (R^2^ > 0.9736 > 0.05) (Figure 1A, Appendix A). To help better understand UA induced cytotoxicity, two concentrations of UA (1.5 µM (Low), 20 µM (IC_50_)) or media only were chosen for evaluation using the Trypan blue cell exclusion assay 48 h post treatment. A significant (*p* < 0.0001) increase in the number of dead cells following treatment with UA IC_50_ was observed when compared to the untreated control (Figure 1b, left panel). When treated with low doses of UA (1.5 μM), there is significant increase (*p* < 0.001) in live cell number compared to cells treated with the UA IC_50_ (Figure 1B, right panel). Interestingly, IC_50_ values calculated for other carcinoma cell lines (A549 and A431) treated with UA were similar to those observed in GBM cells (Appendix A).

Mitochondrial membrane potential (ΔΨm) is an important factor of mitochondrial function and can be an indicator of early intrinsic apoptosis. Collapse of the ΔΨm results in the release of cytochrome C into the cytosol, thus leading to cell death [26]. Loss in ΔΨm was observed following treatment with UA (*p* < 0.05) in a dose dependent manner (Figure 1C, Appendix A), with a significant loss at 25 µM when compared to the untreated control. These data correlate with the loss of respiration function measured using Alamar blue and indicate that the depolarisation of mitochondria is an early feature of UA induced cell death.

### 2.2. UA Demonstrates Enhanced Cytotoxicity Compared to Conventional Chemotherapeutic Drugs

A comparison study was performed between UA and the standard chemotherapeutic drugs used for the treatment of GBM (TMZ) and for recurrent disease (Gefitinib and Carmustine (BCNU)). Little cytotoxicity was observed 48 h after treatment with Gefitinib, TMZ or BCNU, which prevented IC_50_ values being accurately calculated. In contrast, UA demonstrated a significant reduction in cell viability 48 h after treatment (Figure 2A), with an IC_50_ value of 22 µM, similar to that observed in Figure 1a (Appendix A). Cytotoxicity was observed 6 days post treatment using TMZ, BCNU and Gefitinib (IC_50_ values were 28 µM, 79 µM and 16 µM, respectively) in comparison with UA (19 µM) (Figure 2B, Appendix A). Interestingly, over a 6-day period only a small reduction in the IC_50_ value was observed. As individual agents, UA demonstrated greater cytotoxicity over a shorter period and at significantly lower concentrations then that used for TMZ. As demonstrated below in Figure 2C, no additive or synergistic effect was observed between low doses of TMZ and UA. It was postulated that as U-251 MG are TMZ sensitive, UA did not have a demonstrable effect on O-6-Methylguanine-DNA Methyltransferase (MGMT). 

### 2.3. UA Inhibits JNK-Dependent GBM Collective Cell Migration 

One of the most important hallmarks of GBM is the invasive behaviour and ability to migrate into surrounding healthy brain tissue. The scratch assay, also known as the wound healing assay, allows for preliminary quantitative characterisation of collective cell migration and screening of novel therapeutic drugs [27]. The up-regulation of the Jun N-terminal Kinase (JNK) signalling pathway in GBM tumours is commonly associated with tumorigenesis [28] and has previously been implicated in promoting cell migration [29]. Figure 3A demonstrates that treatment of U-251 MG cells with SP600125/UA combination significantly reduced the rate of migration when compared with untreated controls (*p* < 0.05). Migration was reduced to a lesser extent when cells were treated with SP600125 alone. As previously identified above (Figure 1B), no toxicity was observed with 12.5 μM UA after 24 h; however, this may have an effect on migration signalling. Our data demonstrate that 12.5 µM alone significantly reduced the rate of migration compared with the untreated control (*p* < 0.05, Figure 3A). No cumulative effect was observed when UA and SP600125 were combined (*p* > 0.05), indicating a role for UA as an inhibitor of GBM collective cell migration. 

The ability of UA to inhibit cell migration was compared to standard GBM chemotherapeutic drugs. The Pearson correlation coefficient, r, was calculated over the course of 9 h for untreated U-251 MG cells and for cells treated with sub-toxic concentrations of TMZ, BCNU, Gefitinib and UA (Figure 3B). Pearson’s r was strong and negative (i.e., r > −0.95) for both untreated cells and cells treated with TMZ, BCNU and Gefitinib, thus indicating that treatment with these chemotherapeutic agents did not affect the collective cell migration. Pearson’s r for cells treated with UA was positive, indicating that UA had completely blocked migration (Appendix A). Statistical analysis confirmed that a significant level of migration was observed for untreated cells and for cells treated with TMZ, Gefitinib and BCNU (*p* < 0.01, Figure 3B), whereas no significant evidence of migration was observed for cells treated with UA (*p* > 0.05). To further examine the changes in migration following treatment with UA, the rate of scratch closure was calculated. Figure 3C and the associated Appendix A of scratch closure demonstrate that the treatment of GBM cells with UA significantly (*p* < 0.001) reduces the rate of closure of the scratch when compared to the untreated control. No significance was observed for any chemotherapeutic agents when compared to untreated cells. 

### 2.4. UA Induces Caspase-Independent, JNK-Dependent Cell Death

Our data indicate that an alternative mechanism of cell death may be activated following treatment with UA. Three concentrations of UA were tested in order to capture low, moderate and high degrees of cytotoxicity. The JNK specific inhibitor SP600125 alleviated cytotoxicity in U-251 MG cells induced by moderate concentrations of UA (20 µM) (*p* < 0.001), with a significant increase in the IC_50_ value (*p* < 0.05) (Figure 4A). There was no evidence of any inhibitory effects by the broad-spectrum caspase inhibitor zVAD-fmk, as seen in Figure 4B.

Having identified that JNK plays a role in UA induced GBM cell death, it was hypothesized whether stress-activated p38 mitogen-activated protein kinase (p38 MAPK) signalling pathways were involved. However, as demonstrated in Figure 4C, there was no evidence of any inhibitory effects by the p38 kinase inhibitor SB203580. In agreement with previous reports, Figure 4D demonstrated that JNK was involved in TMZ-induced cytotoxicity (*p* < 0.05), in contrast to this, no inhibition was observed when TMZ was combined with zVAD-fmk. These results suggest that UA induces a rapid, caspase- and p38-MAPK independent, JNK dependent mechanism of cell death in U-251 MG GBM cells that involves mitochondrial membrane depolarisation.

### 2.5. UA Triggers the Formation of Lysosomes 

Autophagy is an essential cytoprotective response to pathological stressors, involving the phosphatidylinositol 3-kinase PI3K/AKT/mTOR signalling pathway, and also plays a role in tumorigenesis and invasion [30]. Having identified that UA induces cytotoxic response that is independent of caspases but dependent on JNK signalling, it was hypothesized whether autophagy was activated in response to UA treatment. Following treatment with UA, U-251 MG cells were stained with acridine orange, which is used for the detection of acidic vesicle organelles (AVOs). AVOs are deemed a significant morphological characteristic of the autophagic process [31,32]. As seen in Figure 5A, there is a significant increase in the presence of AVOs measured by quantitative shifts in the FL-2 channel (red fluorescence) intensity ratio following treatment with UA (*p* < 0.005) compared with untreated cells. This suggests the autophagy pathway may be activated during UA-induced cell death.

However, upon further investigation, no evidence of autophagy activation was identified. Furthermore, 3-MA is an inhibitor of autophagosome formation through Class III PI3K activity, which is critical for autophagy activation. As demonstrated in Figure 5B, no evidence of inhibition was observed. No inhibition of cytotoxicity was observed following the addition of U0126 inhibitor (Figure 5C), indicating that UA induced cytotoxicity is independent of Extracellular Signal-Regulated Kinase (ERK) activation. Autophagic flux is used as a measure of autophagic degradation and is associated with late stage autophagy. Cells were treated with chloroquine, a well-known inhibitor which works by destroying autophagosome and lysosome fusion [33]. Chloroquine was unable to prevent cytotoxicity induced by UA (Figure 5D). 

It was postulated whether the acidic vesicles identified in Figure 5A may be an alternative acidic vesicle such as lysosomes. Figure 6A demonstrates a significant increase in acidic vesicles following treatment with UA. However, when stained with lysosomal tracker deep red (Figure 6B), there is also a significant increase in fluorescence intensity following UA treatment (*p* < 0.0001). This provides evidence that the vesicles detected using (Acridine Orange) AO are lysosomes. To confirm the presence of lysosomes, 3D isosurface rendered z-stack confocal images were generated from cells treated with both AO and LysoTracker Deep Red. Figure 6C demonstrates co-localisation, as depicted in blue in the merged panel. As previously described, AO also binds with single-stranded nucleic acids and emits orange fluorescents, which can be seen in both the merged panel of the untreated control and UA treated cells [34].

## 3. Discussion

An estimated 25% of chemotherapeutic drugs used during the last 20 years are directly derived from plants, while another 25% are chemically altered from natural products. It is noteworthy that only 5–15% of the approximately 250,000 higher plants have ever been investigated for bioactive compounds [35]. UA, found in the leaves and fruits of many plants, induces cytotoxicity in cancer cells while remaining non-toxic to normal cells [11,36]. Derivatives of UA have also demonstrated increased antitumor efficacy in vivo [37,38,39]. This paper demonstrates that UA induces JNK-dependent and caspase independent cytotoxicity in a human GBM model. Other groups have reported various cytotoxic and pro-survival signalling pathways activated by UA in GBM cancer cell lines, including apoptosis, autophagy, senescence and necrosis. The variability in responses observed appears partly dependent on the treatment regimen, relative resistance to TMZ and the GBM model used. Lower doses of UA were generally anti-proliferative in GBM cells without directly inducing cytotoxicity [21,22]. Cytotoxicity observed at higher doses of UA does not always carry the hallmark indicators of apoptosis. Biochemical indicators of apoptosis, including caspase activation, phosphatidylserine exposure and DNA fragmentation, were evident when TMZ-sensitive GBM cells, such as C6 cells [14] and U251 cells [13], were treated with UA in vitro. However, treatment of TMZ-resistant GBM cells with UA led to cytotoxicity associated with caspase-independent signalling, such as necrosis in DBTRG-05MG [12], LN18 and T98G cells [20]. Interestingly, in TMZ-sensitive U87MG cells, autophagy via Reactive Oxygen Species (ROS)-induced Endoplasmic Reticulum (ER) stress was reported in parallel with cytotoxicity [24]. TMZ resistance is believed to be conferred by MGMT expression in GBM cells and correlates with poor prognosis [40]. Our finding that UA induces JNK-dependent, caspase independent cytotoxicity in U-251 MG cells differs to that described in other TMZ-sensitive cell lines. UA was shown to reduce the expression of MGMT in MGMT-expressing cells and induce synergistic cytotoxicity in TMZ-resistant GBM cell lines T98G, LN18 and LN229 when co-incubated with TMZ. Moreover, the rate of tumour growth was reduced when mice challenged with flank LN18 xenograft tumours were treated with UA and TMZ [20]. We did not observe any additive or synergistic effect when U-251 MG cells were treated with a combination of TMZ and UA. This may be due to the steep Hill slope observed when U-251 MG cells were treated with UA, compared to T98G, LN18 and LN229 cells that display significant resistance to UA and TMZ, or perhaps MGMT expression in U-251 MG cells is regulated through a different UA-insensitive signal transduction pathway.

Cell migration is an important feature in many biological processes, such as tissue inflammation and tissue homeostasis, and plays a central role in cancer progression [27]. GBM is highly invasive, resulting in tumour progression and poor prognosis. Collective cell migration has been observed in many tumour types, i.e., breast [41,42]. Recent studies have also demonstrated that GBM cells and GBM cancer stem cells can migrate as both a single cell and as a collective, which is thought to influence their ability to infiltrate the surrounding tissue [43,44]. Therefore, it is necessary to identify novel bioactive compounds that demonstrate both anticancer and anti-migratory properties. UA has been implicated in regulating cell migration and invasion in gastric cancer cells, breast and lung cancer cells [45,46,47]. Oleanolic acid, a pentacyclic triterpenoid, which is structurally similar to UA differing only by the position of one methyl group on the ring E [48], has also demonstrated anti-migratory and anti-invasive effects in GBM cells by inactivating the MAPK/ERK signalling pathway [49]. Using the scratch assay, our data demonstrate that UA is a potent inhibitor of U-251 MG collective cell migration. Similarly, an investigation in human breast cancer cells found that migration was also affected through JNK, Akt and mTOR signalling following treatment with UA [46]. Interestingly, while we observed that the inhibition of JNK reduced collective cell migration, we did not observe any evidence that JNK was directly involved in the inhibition of migration following treatment with UA. The contrary appears to be the case; rather than suppression of JNK by UA, as reported by Yeh and colleagues, our data indicate that JNK activity is promoted by UA in GBM cells and mediates cytotoxicity. No adverse interaction between UA and the JNK inhibitor was observed, suggesting that UA likely affects GBM migration using a JNK-independent signalling pathway in GBM cells. Moreover, sub-toxic doses of UA were potent inhibitors of GBM collective cell migration, whereas TMZ, Gefitinib or BCNU were unable to significantly reduce collective cell migration. 

Several competing hypotheses have been put forward regarding the signalling pathways responsible for UA induced cytotoxicity, with both caspases and kinases being implicated [13,50,51,52,53]. Studies to date using GBM cells suggest that caspase involvement in UA induced cell death appears to be associated with TMZ sensitivity, and observed caspase independent cell death was described as necrosis. Our data suggest that caspases are not required for UA-induced cytotoxicity in U-251 MG cells. Alternatively, our data demonstrate that JNK regulates cytotoxicity in GBM cells. No hallmarks of necrosis or necroptosis were observed, such as cell swelling and lysis, suggesting that caspase-independent apoptosis is induced by JNK in U-251 MG cells in response to UA. JNK-dependent apoptosis involving caspase activation has previously been reported for various human cancer cell lines treated with UA, including pancreatic [54], bladder [55] and prostate [52]. We found that UA induced JNK-dependent, caspase-independent cell death correlates with findings observed in human HCT15 colorectal cancer cells [53]. Interestingly, we found that SP600125 could significantly lower cell death at a concentration close to the IC_50_ of UA (i.e., 20 µM), but not at higher doses. Our observations correlate with those observed by Zhang et al. in prostate cancer cells [52] and may indicate that other cytotoxic signalling pathways can be activated by UA, depending on the cell line. 

JNK activation has also been implicated during the activation of autophagy by UA. Xavier et al. reported that UA increased expression of the autophagic indicators LC3-II and P62 in colorectal carcinoma cells, which were inhibited using SP600125 [53]. Similar findings of ER stress-induced JNK activation and autophagy were reported in U-87MG cells [24]. In both studies, autophagy was associated with cell death. Similarly, Kung et al. observed UA-induced cell death without any DNA fragmentation, caspase activation or phosphatidylserine flipping in human cervical carcinoma cells, concluding that autophagy is involved in promoting UA cytotoxicity [56]. The role of autophagy in the cell death process is not fully understood. The term “autophagic cell death” was widely used in the literature, based on the observation that autophagy is commonly associated with cell stress events, and that instances of cell death accompanied by a massive cytoplasmic vacuolization lead to the conclusion that autophagy is an enabler of cell death [57]. However, recently it was noted that inhibition of the processes that are essential for cell death often does not result in inhibition of the cells’ demise, but more often results in alternative biochemical pathways inducing cell death via a different mechanism [58]. 

Having identified the presence of AVOs by acridine orange staining, it was necessary to elucidate their identify and role in UA induced cell death. It is important to note that acridine orange cannot be used as an identifier of autophagic vesicles, but more as an indicator. Acridine orange crosses into acidic compartments and becomes protonated. Therefore, it cannot be ruled out that the observed shift in fluorescence is from the uptake of acridine orange by other acidic vesicles, such as lysosomes, and not autolysosomes. Our data demonstrated no inhibition of autophagy with the PI3 Kinase inhibitor 3-MA. Leng et al. observed a cytoprotective effect using Wortmannin (a PI3K inhibitor) and ATG5 small inhibitory RNA (siRNA) in combination with UA. However, they also observed no cytoprotective effect using PI3K inhibitor 3-MA, despite observing inhibition of LC3-II and accumulation with LY294002. Our observation that the inhibition of PI3-K using 3-MA did not alleviate the cytotoxic effects of UA correlates with Leng et al. [56]. To further confirm that autophagy was not playing a role, we observed no inhibitory effects of chloroquine on UA treated cells. Chloroquine is a lysosomotropic agent that prevents endosomal acidification and therefore inhibits autophagy as it raises the lysosomal pH, leading to the inhibition of both the fusion of autophagosome with lysosome and lysosomal protein degradation. It was then postulated that the observed influx in acidic vesicle organelles was lysosomes, thus providing a rationale as to why no evidence of autophagy was identified. In recent years, lysosome permeabilization has been associated with cell death [25,59,60]. Recent data have demonstrated that another pentacyclic triterpenoid, Betulinic acid (BA), induces both mitochondrial and lysosomal dysfunction, which in turn resulted in autophagic flux inhibition [61]. Similar to our data, Martins et al. demonstrated an inhibition of cell death following co-treatment with chloroquine and BA. The authors identify that BA mediated lysosomal damage was capable of comprising autophagy, and perhaps UA is having a similar effect on lysosomes in U-251 MG treated cells. In addition, a similar effect was also observed following exposure of NH2-PS nanoparticles to mouse embryonic fibroblasts, where after an 8 hr exposure, autophagic flux was blocked due to damage to the lysosomes resulting in LDCD [59]. Interestingly, Martins et al. also demonstrated that cell death following BA was associated with both lysosome damage and mitochondrial depolarisation. In this study, our data show rapid depolarisation of the mitochondria following treatment with UA. It is possible that due to the steep cytotoxic Hill slope, the rapid depolarisation of the mitochondria and clear evidence of lysosome accumulation, UA induces physiological stress, to the point that autophagy was unable to return to and maintain homeostasis, therefore resulting in lysosome associated cell death. 

Together, this study demonstrates that UA has a greater capacity to induce cell death and inhibit GBM collective cell migration over currently used chemotherapeutic agents, demonstrating efficacy as a potential therapeutic target. Based on our findings that UA induces a JNK regulated form of cell death that results in the rapid depolarisation of the mitochondria and the accumulation of lysosomes, it is plausible that LDCD may be playing a role. However, further investigation is required to fully elucidate whether lysosomes are damaged following treatment with UA, similarly to that observed with other triterpenoids.

## 4. Materials and Methods 

### 4.1. Cell Culture 

Human glioblastoma U-251 MG, formerly known as U-373 MG (ECACC 09063001)), cells were obtained from Dr Michael Carty (Trinity College Dublin). U-251 MG cells were cultured in Dulbecco’s Modified Eagle Medium (DMEM) (Sigma-Aldrich, Arklow, Ireland) supplemented with 10% Fetal bovine serum (FBS) (Sigma-Aldrich, Arklow, Ireland) and were maintained in a humidified incubator containing 5% CO_2_ at 37 °C. Media were changed every 2–3 days until 80% confluency was reached. Cells were routinely sub-cultured using a final 1:1 ratio of 0.25% trypsin (Sigma-Aldrich, Arklow, Ireland) and 0.1% Ethylenediaminetetraacetic acid (EDTA) (Sigma-Aldrich, Arklow, Ireland).

### 4.2. Cytotoxicity 

Dose response curves for commonly employed chemotherapeutic drugs used for the treatment of GBM: Temozolomide (TMZ) (Sigma-Aldrich, Arklow, Ireland), Carmustine (BCNU) (Sigma-Aldrich, Arklow, Ireland) and Gefitinib (Insight Biotechnology ltd, Wembley, UK) UA standard (Sigma-Aldrich, Arklow, Ireland) were established. UA standard, TMZ and Gefitinib were dissolved in Dimethyl sulfoxide (DMSO) (Sigma-Aldrich, Arklow, Ireland) and BCNU in sterile H_2_O and stored at −20 °C. These stocks were subsequently used to make the working standard solutions in media. The highest concentration of DMSO used was 0.5%. U-251 MG cells were seeded at a density of 1 × 10^4^ (24 and 48 hr exposure time points) in 96 well plates (Sigma-Aldrich, Arklow, Ireland) with 100 µL media per well. In order to establish accurate IC_50_ values for the chemotherapeutic compounds, a lower seeding density of 2.5 × 10^3^ and a 6-day exposure period were required. Plates were left overnight in the incubator at 37 °C with 5% CO_2_ to allow the cells to adhere. Existing media were removed from each well and cells were treated with either a chemotherapeutic drug, UA or solvent control (0.5% DMSO) and incubated for the appropriate time point. No deleterious effects were observed from the control solvent. 

### 4.3. Cell Viability Assays

Trypan blue cell exclusion assay was performed as an initial evaluation of cell health following treatment with UA. Cell were trypsinized as described above. A Trypan Blue (Fischer Scientific, Ballycoolin, Ireland) cell suspension was counted using a haemocytometer as per manufacturer’s instructions. Cell viability was also measured biochemically using the Alamar Blue assay (Fischer Scientific, Ballycoolin, Ireland). Alamar blue is an oxidation-reduction (redox) fluorogenic indicator of cellular metabolic reduction. After each exposure time point, (24 h or 48 h) cells were washed once with sterile Phosphate-buffered saline (PBS). A 10% Alamar blue solution in the DMEM was added to each well and incubated at 37 °C for 2.5 h. Fluorescence was read at an excitation wavelength of 530 nm and an emission wavelength of 595 nm using the Victor 3V 1420 (Perkin Elmer) multi-plate reader.

### 4.4. Flow Cytometry 

#### 4.4.1. JC-1 Mitochondrial Membrane Potential Assay

As described above, cells were plated and exposed to varying concentrations of UA for 48 h. Cells were then harvested and stained with 10 µg/mL JC-1 dye, a mitochondrial membrane potential probe (Biosciences, Dublin, Ireland) (Galluzzi et al., 2007), at room temperature for 10 min and analysed by flow cytometry (BD Accuri C6). JC-1 was excited using the argon laser at a wavelength of 488 nm. Fluorescence was measured using the FL1 (530 nm) and FL2 (585 nm) channels with emission spectral overlap compensation (7.5% FL1/FL2 and 15% FL2/FL1). 

#### 4.4.2. Acridine Orange (AO)

Cells were harvested and stained with 1 µg/mL AO and incubated at 37 °C for 20 min. Cells were then washed twice with sterile PBS and analysed by flow cytometry (BD Accuri C6) with an excitation of 475 nm and emission 590 nm. Detection of AVOs was achieved using the FL1 (green) vs. FL2 (orange) channels and compensation was set at approximately 7% removing FL2 signal from FL1 and approximately 16% removing FL1 signal from FL2 for each plot.

### 4.5. Scratch Assay

The scratch assay allows for the preliminary examination of the effects of TMZ, Gefitinib, BCNU, and UA on the migration of U-251 MG cells. Cells were treated with each compound, at a sub-toxic concentration, to prevent cytotoxic responses but potentially inhibit migration. U-251 MG cells were seeded at 0.9 × 10^6^ cells in individual 35 mm dishes and incubated for 24 h. A scratch was performed in each dish prior to treatment using a 200 µL sterile pipette tip. Utilising the data observed from the UA dose response curve, a sub-toxic concentration (12 μM) was chosen. Cells were treated with either media alone, DMSO (0.1%), or 12.5 μM UA, each scratch was examined under a light microscope and images were taken using the (Nikon Eclipse 700). Multiple images were taken for each time point and the average size of scratch for that time point was obtained. Image analysis was performed using image processing and analysis software, Image J [62]. 

### 4.6. Inhibitor Studies

U-251 MG Cells were plated in 96 well plates as described above and left to adhere overnight. Cells were pre-treated for 1 h with either zVAD-FMK; caspase inhibitor (BD Bioscience, Oxford, England), SP600125; JNK inhibitor, SB203580; p38 MAP Kinase Inhibitor (Sigma Aldrich, Arklow, Ireland), U0126; MEK1 and MEK2 Kinase Inhibitor (Sigma Aldrich, Arklow, Ireland), chloroquine or 3-methyladenine (3-MA), autophagy inhibitors (Sigma Aldrich, Arklow, Ireland) after which the UA standard (IC50) was added to each well. Cell viability was assessed 48 h later using Alamar Blue cell viability assay. 

### 4.7. Confocal Microscopy 

U-251 MG cells were plated in 35 mm glass bottom dishes (MatTek Corporation, Ashland Massachusetts, USA) at 1 × 10^5^ cells per ml and incubated for 24 h. Cells were treated with 20 µM UA for 48 h. After treatment, cells were loaded with AO (1 µg/mL for 15 min at 37 °C) and LysoTracker Deep Red (Thermo Fisher Scientific) (50 nM for 30 min at 37 °C). Images were captured on a Zeiss 510 LZSM confocal inverted microscope. The corresponding filter settings were as follows: AO, excitation 477 nm, emission 585–615 nm; LysoTracker Deep Red, excitation 633 nm, emission 649–799 nm. All images were taken using live cells. The total integrated density of fluorescence of each cell was quantified using ImageJ (v1.49, NIH) software. The quantified integrated density equals to the sum of the pixel values in the selected fluorescent area.

### 4.8. Isosurface Rendering

Representative confocal Z-scans of U-251 MG cells were processed for a three-dimensional reconstruction and visualization of volumetric co-localization of the acidic vesicles (ACRID) and lysosomes (LYSO). Fluorescent ACRID-LYSO colocalizing voxels were detected by a computerized software using Coloc module (Imaris Bitplane, South Windsor, CT, USA), then a new channel with these voxels was extracted. Then, this colocalizing voxels channel was used to generate a three-dimensional isosurface employing Surface module (Imaris BitPlane, South Windsor, CT, USA) with the appropriate threshold and resolution. Similarly, ACRID and LYSO channels were also used to generate three-dimensional renderings as isosurfaces. Merging of the volumetric isosurfaces was set to illustrate the degree of colocalizing objects in control and treated glioma cells. Adequate shadowing and angle of three-dimensional rotation was applied to show the relevant cellular structures.

### 4.9. Statistical Analysis 

All experiments were performed in triplicate, independently of each other with a minimum of five replicates per experiment. Data shown are pooled and presented as mean ± SEM (*n*= total number of replicates) unless stated otherwise. Statistical analysis was performed using Prism 5, GraphPad Software, Inc. (San Diego, CA, USA). Unless otherwise indicated, differences were considered significant with a * *p* value < 0.05.

## Figures and Tables

**Figure 1 pharmaceuticals-14-00091-f001:**
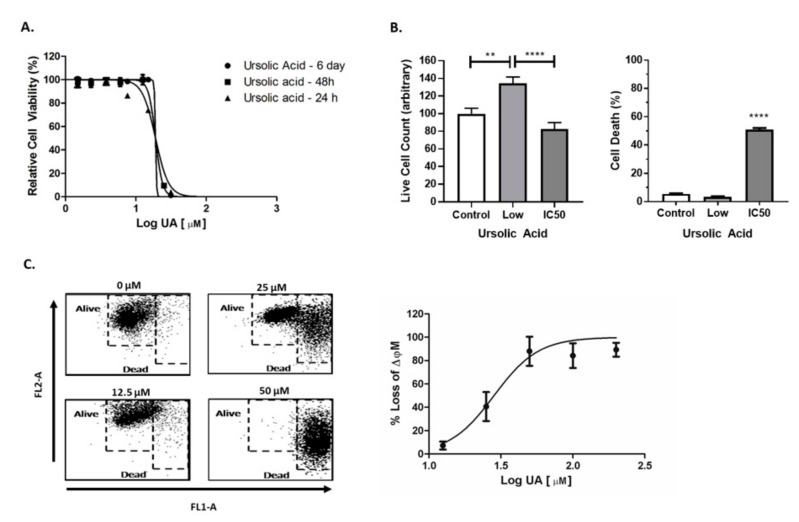
Ursolic acid (UA) induces mitochondrial membrane depolarisation. (**A**) U-251 MG cells were exposed to increasing concentrations of UA. Cell viability was assessed using Alamar blue at 24 h, 48 h and 6 days (*n* = 3). (**B**) Cells were treated with 1.5 µM UA (Low), 20 µM (IC50) or media only (Control) for 48 h. Viability was measured using the Trypan blue cell exclusion assay (** *p* < 0.005; **** *p* < 0.001); *n* = 3) (**C**) After a 48 hr exposure to UA, cells were loaded with 10 µg/mL JC-1 dye and analysed by flow cytometry. Data shown depict cell death measured by quantitative shifts in the ΔΨm (red to green) fluorescence intensity ratio with increasing concentrations of UA (*n* = 3). Data shown were normalised to the untreated control and are shown as the % mean ± SEM (standard error of the mean). Statistical analysis was carried out using non-linear regression.

**Figure 2 pharmaceuticals-14-00091-f002:**
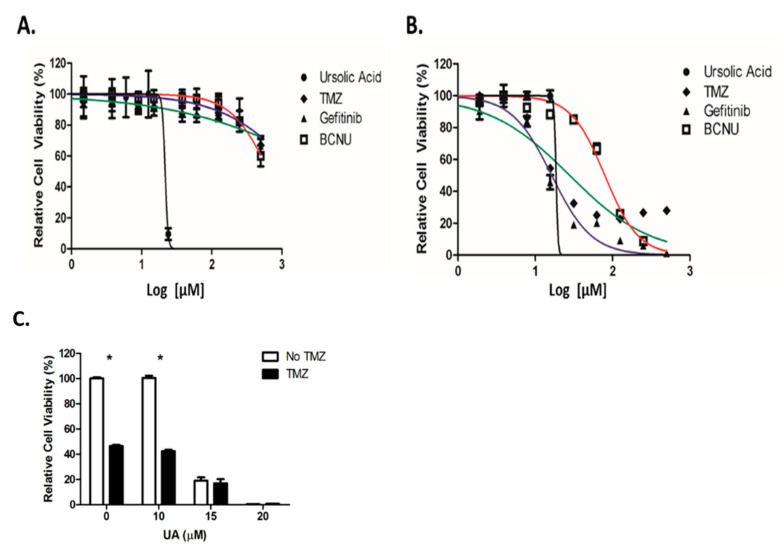
UA exhibits increased cytotoxicity over conventional chemotherapeutics. (**A**,**B**) U-251 MG cells were treated with increasing concentrations of UA (0–200 μM), Temozolomide (TMZ), BCNU, or Gefitinib (0–500 μM) for 48 h (**A**) or 6 days (**B**) and analysed using Alamar blue. Statistical analysis was carried out using non-linear regression analysis and Two-way ANOVA with Bonferroni post-tests, (*n* = 3) (* *p* < 0.001). (**C**) Cells were treated with 0–20 μM UA after which 15 μM TMZ was added, incubated for 6 days and analysed by Alamar blue. Data shown were normalised to the untreated control and are shown as the % mean ± S.E.M. Statistical analysis was carried out using Two-Way ANOVA with Bonferroni post-test (* *p* < 0.001) (*n* = 3).

**Figure 3 pharmaceuticals-14-00091-f003:**
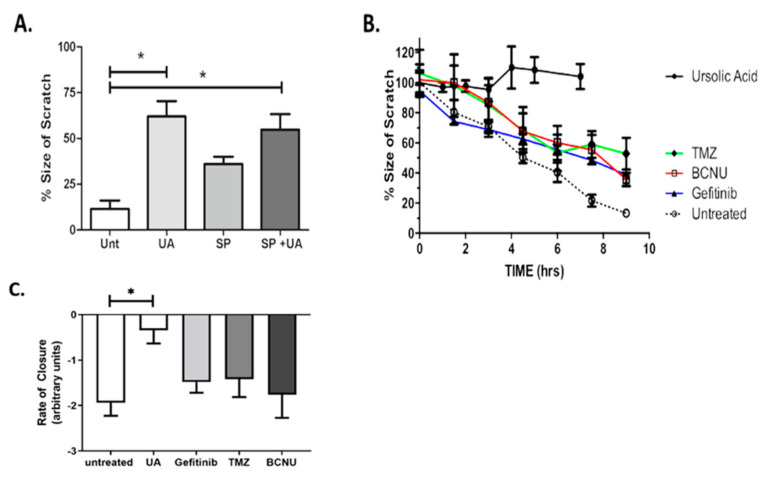
UA inhibits JNK-dependent Glioblastoma multiforme (GBM) collective cell migration. (**A**) Cells were treated with either 12.5 μM UA, 12.5 μM SP600125 or UA/SP combined. Images were taken at the time of the scratch and 7 h later. Statistical analysis was carried out using One-Way ANOVA with Tukey’s multiple comparison post-test (* *p* < 0.05) (*n* = 3). (**B**,**C**) Cells were treated with TMZ (100 μM), Gefitinib (25 μM), BCNU (50 μM), UA (12 μM) or were untreated. Time dependent closure of the scratch was analysed over a 9 h period. Statistical analysis was carried out using linear regression analysis and Two-Way ANOVA with Bonferroni post-tests. The rate of closure was determined by One-Way ANOVA with Tukey’s test for multiple comparisons (*n* = 3). Image analysis was performed using ImageJ software.

**Figure 4 pharmaceuticals-14-00091-f004:**
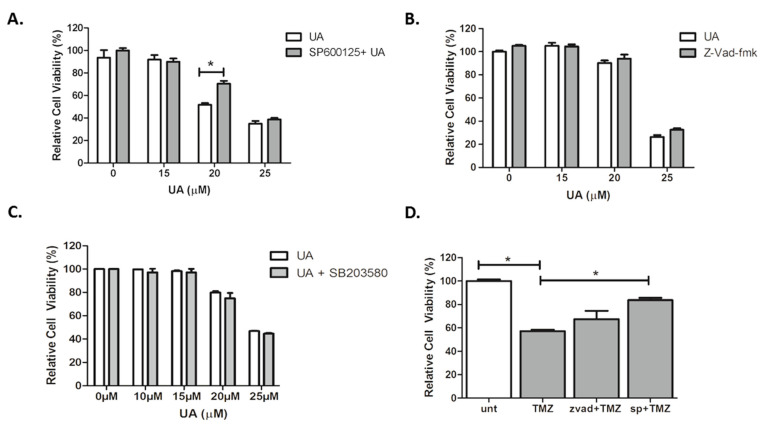
UA induces caspase-independent, JNK-dependent cell death. (**A**–**C**) Cells were pre-treated with either 12.5 µM SP600125, 50 µM zVAD-FMK or 10 µM SB203580 for 1 h. Increasing concentrations of UA were added for 48 h and analysed using Alamar blue. Statistical analysis was carried out using Two-Way ANOVA with Bonferroni post-test. (**D**) Cells were pre-treated with either zVAD-fmk (50 μM) or SP600125 (12.5 μM) for 1 h prior to addition of TMZ (26.5 μM) for 6 days. Cell viability was analysed by Alamar blue. Statistical analysis was carried out using One-Way ANOVA with Tukey’s post-test (* *p* < 0.05). All experiments were normalised to untreated control and expressed as a % of the SEM (*n* = 3).

**Figure 5 pharmaceuticals-14-00091-f005:**
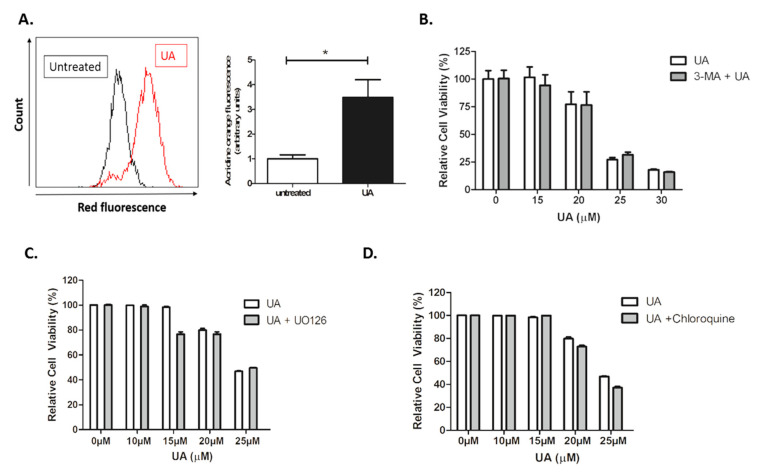
UA initiates formation of acidic vesicles but no evidence of autophagy. (**A**) U-251 MG cells were treated with 22 µM UA. After 48 h, cells were loaded with 1 µg/mL acridine orange fluorescent probe and analysed by flow cytometry. Data shown were normalised to the untreated control and represented as arbitrary units ± S.E.M (*n* = 3). Quantification of mean fluorescence index and statistical analysis was performed using non-parametric t-tests (* *p* < 0.05). (**B**–**D**) U-251 MG cells were pre-treated with 5 mM 3-Methyladenine (3-MA), 10 µM U0126, or 25 µM Chloroquine for 1 h prior to addition of UA. Cells were incubated for 48 h and analysed by Alamar blue. Data shown were normalised to the untreated control and are shown as the % mean ± S.E.M (*n* = 3). Statistical analysis was carried out using Two-Way ANOVA with Bonferroni post-test.

**Figure 6 pharmaceuticals-14-00091-f006:**
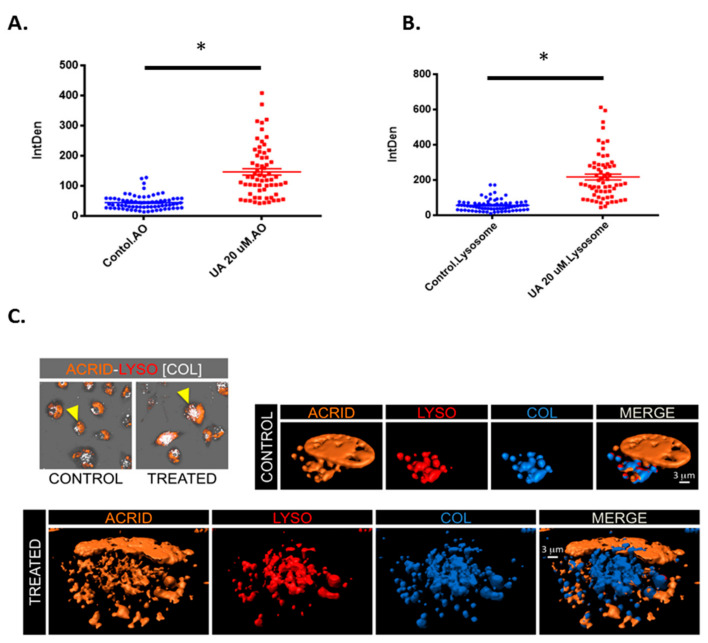
UA triggers lysosome accumulation. (**A**) U-251 MG cells were exposed to 20 µM UA. After 48 h, cells were loaded with acridine orange (1 µg/mL for 15 min at 37 °C) and (**B**) 50 nM lysotracker (50 nM for 30 min at 37 °C) and analysed by confocal microscopy. The fluorescence intensity of AO green and orange channel, and LysoTracker Deep Red was quantified using ImageJ software and compared to the untreated control. Statistical analysis was carried out using an unpaired t-test (* *p* < 0.0001). (**C**) Images showing co-localisation analysis of LysoTracker Deep Red (LYSO) and AO (ACRID) channels as white voxels (COL) for both control and UA treated cells. Three-dimensional visualization of control and treated cells demonstrates the co-localisation of orange (acridine orange) and red (LysoTracker Deep Red) indicated with a blue isosurface (COL) together with rendered isosurface of the AO (ACRID) and LysoTracker (LYSO) in separated channels or merged images (MERGE).

## Data Availability

The authors will freely release all data underlying the published paper upon direct request to the corresponding author.

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
