# Peer review of "Ursolic Acid Inhibits Collective Cell Migration and Promotes JNK-Dependent Lysosomal Associated Cell Death in Glioblastoma Multiforme Cells"

_pharmaceuticals, 2021, doi:10.3390/ph14020091_

Round 1

Reviewer 1 Report

Dear Dr. Gillian,

I have read this interesting piece of research. I would like to appreciate the quality of work and the presentation of meaningful data.

1. In the overall discussion part, there is a need to do proper punctuation.
2. from the last paragraph line no. 492-499, the text is in bold format, which needs to be unbold (i am not sure if it was submitted in this form originally)
3. Title MATERIAL AND METHODS need to be shifted in a new line.
4. If there is possibility, then graphical presentation of the possible mechanism of action of UA is supposed to be more effective to elaborate.

Best regards.

Reviewer 2 Report

The paper of Gillian E. Conway, Deimante Zizyte, Julie Rose at al. "Ursolic acid inhibits collective cell migration and promotes JNK-dependent lysosomal associated cell death in glioblastoma multiforme cells" is devoted to the study the cytotoxic properties and mechanism of action of Ursolic acid (UA), the well-known natural triterpene acid widely distributed in the plant kingdom. The relationship between cell death and migration was studied by using different assays. It was found that UA induced a dose dependent cytotoxic response, and a caspase independent, JNK dependent mechanism of cell dearh. Migration studies establishrd that UA inhibited CBM collective cell migration independently of the JNK signalling pathway. It was concluded that UA rapidly induced a lysosomal associated mechanism of cell death, and acting as an inhibitor of CBM collective cell migration. Results are interesting for development of anticancer agents based on natural products, but the manuscript should be revised to improve it. There are some expanded parts (see the yellow color in the text) (it is similar to a review) that may be written in more short form. Abstract should be revised too to delete no need details of experiments. There are a lot of diagrams, some data it is better present as tables. This paper is more suitable for medical chemistry or oncology journals such as J. Neurooncol., Oncotarget, Cancer Lett., Biochim. Pharmacol. etc. 

Reviewer 3 Report

It is a very good job/manuscript with well-planned experiments. The presentation of the results does not raise any objections. The subject of the manuscript is interesting because it concerns the possibility of using a natural substance - ursolic acid in anti-cancer therapy of glioblastoma multiforme.

Reviewer 4 Report

The work deals with an interesting and topical topic. It's well-focused.

 The introduction helps perfectly to understand the reasons that have led the research team to raise the overall objective of the study. The specific objectives of the investigation should be realized in line 104. That way the rest of the chapters become clearer.

Chapter 2. Results need to be improved in terms of presentation. The feet of the figures are too long, and the graphics too small, especially in the cases of Figure 1 (paragraph C) and Figure 4 (paragraph C). In general, the quality of the graphics is improvable.

The sub-assessments of the Results should be modified in their wording so that what data (results) are not confused with the interpretation of the data (discussion). The discussion section should bear the same sections as Results so that we can easily assess the extent to which the statements in the discussion are based on the data obtained.

Table 1 is too large for the information it provides. It can be omitted and included in the text as redacted phrases.

In Material and Methods, there are typography errors (lines 499 and earlier; line 549).

A list of abbreviations is missing.

In short, I believe that the work may be admitted to publication in Pharmaceuticals, but the modifications I suggest should be made in advance, and afterwards, it should be reviewed, perhaps by a more specialized author on the specific subject on which the research is carried out.

Round 2

Reviewer 2 Report

The paper of G. E. Conway, D. Zizyte, J. Rose, M. Mondala, Z. He, L. Lynam, M. Lecourt, C. Barcia, O. Howe, J.F. Curtin "Ursolic acid inhibits collective cell migration and promotes JNK-dependent lysosomal associated cell death in glioblastoma multiforme cells" was corrected and the text was reduced. It may be accepted for publication. 

Reviewer 

Author Response

Reviewer's comments:

The paper of G. E. Conway, D. Zizyte, J. Rose, M. Mondala, Z. He, L. Lynam, M. Lecourt, C. Barcia, O. Howe, J.F. Curtin "Ursolic acid inhibits collective cell migration and promotes JNK-dependent lysosomal associated cell death in glioblastoma multiforme cells" was corrected and the text was reduced. It may be accepted for publication.

Author's response:

We would to thanks the review for their detailed review of our manuscript.  

Reviewer 4 Report

The authors wrote in the response coverletter that they have modified and written in colours the parts suggested by 4 Reviewers, but I am afraid I did not find the blue ones (Reviewer 3). Anyway in my case all the modifications have been attended so I think the paper can be admitted for publication. 

Please review the fonts at Supplementary Material, and homogenize them. 

Author Response

Reviewer comments:

The authors wrote in the response cover letter that they have modified and written in colours the parts suggested by 4 Reviewers, but I am afraid I did not find the blue ones (Reviewer 3). Anyway in my case all the modifications have been attended so I think the paper can be admitted for publication. 

Please review the fonts at Supplementary Material, and homogenize them. 

Author response :

Dear Reviewer,

We would like to thank you for your thorough review of our manuscript. Apologies for the confusion, as reviewer 3 had no comments there was no blue text in the manuscript. As per your suggestion we have amended the font in the supplemental document.